# Modular Site-Specific Conjugation of Nanobodies Using Two Co-Associating Tags

**DOI:** 10.3390/ijms232214405

**Published:** 2022-11-19

**Authors:** Eric Moeglin, Lina Barret, Bruno Chatton, Mariel Donzeau

**Affiliations:** 1UMR7242 Biotechnologie et Signalisation Cellulaire, Université de Strasbourg, F-67412 Illkirch, France; 2Nanotranslational Laboratory, ICANS, Centre Paul Strauss, 3 Rue de la Porte de l’Hôpital, F-67000 Strasbourg, France; 3IMPReSs Facility, Biotechnology and Cell Signaling, CNRS, University of Strasbourg, F-67412 Illkirch, France

**Keywords:** V_H_H, nanobody, conjugation, site-specific

## Abstract

The homogeneous labeling of antibodies and their fragments is a critical step for the generation of robust probes used in immuno-detection applications. To date, numerous chemical, genetic and peptide-based site-specific coupling methods have been developed. Among these methods, co-assembling peptide-tags is one of the most straightforward and versatile solutions. Here, we describe site-specific labeling of nanobodies through the use of two co-associating peptides tags, E3 and K3, originating from the tetramerization domain of p53. These E3 and K3-tags provide a simple and robust method for associating stoichiometric amount of V_H_H and fluorescent probes, either fluorescent proteins or fluorochromes, at specific positions. As a proof of concept, a nanobody targeting the human epidermal growth factor receptor 2 (HER2), the nano-HER2 was genetically fused to the E3 and associated with different fluorescent K3-derivates. Entities were produced separately in *Escherichia coli* in soluble forms at high yields and co-assembled in vitro. These molecular probes present high binding specificity on HER2-overexpressing cells in flow-cytometry with relative binding constants in the low nanomolar range and are stable enough to stain HER2-receptor on living cells followed detection using fluorescent confocal microscopy. Altogether, our results demonstrate that the non-covalent conjugation method using these two co-associating peptides can be easily implemented for the modular engineering of molecular probes for cell immuno-staining.

## 1. Introduction

Antibodies and their derived fragments are essential tools to visualize cellular proteins and cell surface receptors. To this regard, labeling of these binders is a critical step for the generation of high quality and homogenous probes suitable for various applications such as fluorescence-microscopy, flow cytometry, diagnostic, therapeutic applications and molecular imaging. A very promising alternative to antibodies, Fabs and scFvs is the use of variable domains of heavy chain antibodies from Camelidae (V_H_Hs or nanobodies). Indeed, V_H_Hs are highly soluble, stable under various physicochemical conditions and can be produced at high yield in *E. coli.* They can also be easily engineered, opening a wide range of labeling strategies [1,2,3,4].

V_H_Hs can be labeled either covalently or non-covalently with fluorochromes or fluorescent proteins [5]. However, each of these methods suffers from several drawbacks. For instance, chemical methods involving the covalent conjugation to either side chain ε-amine group of a lysine or sulfhydryl group of a cysteine may alter residues inside or near the antigen-binding site, leading to the generation of non-functional reagents. Maleimide conjugation to cysteine residues can lead to labeling heterogeneity and in some cases to irreversible unfolding, especially if buried scaffold cysteine residues are conjugated, resulting in a heterogeneous population of labeled tracer [5]. Therefore, several alternatives have been developed to circumvent these shortcomings. Thus, introduction of an additional unpaired cysteine at the carboxyl-terminal end of the nanobody for site-specific conjugation via a thio-ether bond has proven to be a reliable method for the generation of homogenous tracers [6]. This approach has been further developed on an anti-Nup98 nanobody in which cysteine residues were engineered at the nanobody surface for site-specific conjugation, leaving the internal framework cysteine residue fully intact [5]. But to better guide cysteine assignment in the nanobody framework, the crystal structure of the anti-Nup98 nanobody in complex with the Nup98 protein had to be solved, making this approach cumbersome for its application on different nanobodies.

Apart from the genetic engineering of unpaired cysteine residue for site-specific labeling, other alternatives explored peptide-tag approaches for site-specific protein labeling. Enzyme-based tags provide high specificity and enable fast covalent site-specific labeling reactions [7]. Peptide-based recognition tags are categorized into three groups. First, the chemo-enzymatic labeling techniques, such as biotin ligase, transglutaminase or Sortase A, can achieve site-specific labeling of a protein of interest in one or two coupling steps [7]. For example, a short tubulin-derived recognition sequence (Tub-tag) was used successfully for C-terminal labeling of nanobodies [8]. Moreover, the protein-ligation capacity of the enzyme Sortase A allowed site-specific coupling of three different probes to an anti-HER2 nanobody for in vivo imaging applications [3]. However, to achieve high labeling yields, long reaction times and high concentrations in the µM range of substrates were required. For the Sortase A reaction, nucleophilic probes have to be synthetized, and the overall efficiency of the conjugation procedure does not exceed 30 to 50% [3]. The second group of peptide-based tags is set on metal ion-dependent labeling with cysteine-rich tags or metal-chelating tags such as His-tags, allowing one-step labeling. However, cytotoxicity, background signals due to non-specific interactions with other proteins and non-specific binding are the main drawbacks [7]. The last group consists of peptide-binding peptide tags. One example is the artificial heterodimeric coiled-coil peptides [9]. These tags, which form heterodimers with high affinity in the nM range, were, however, not tested for site-specific labeling of V_H_Hs [10].

Beside chemical labeling and peptide tags, a vast number of fluorescent proteins (FPs) with a broad color palette spanning nearly all visible and far-red spectral regions are available [11]. Nanobodies genetically fused to these fluorescent encoded probes are called chromobodies and have been successfully expressed in living cells to trace intracellular proteins using fluorescence microscopy, such as Dmnt1 or microtubules [12,13]. To date, one anti-CD38 chromobody genetically fused either to eGFP or to mCherry has been used successfully for flow cytometry [14].

In this report, we describe a straightforward modular strategy for a site-specific conjugation of nanobodies suitable to either fluorochromes or FPs using the co-associating peptides E3 and K3 [15,16,17,18]. These two short peptidic sequences of 31 amino acid residues, which originate from the tetramerization domain of p53 (residues 325–355), co-associate exclusively through helix–helix contacts to form a dimer of dimers. The α-helical charged interface involving lysine and acid glutamic residues modulates tetramer stability through salt bridges. We describe the use of E3 and K3 peptides to engineer bifunctional site-specific labeled macromolecules for immuno-detection of receptor proteins in fluorescence microscopy and in flow cytometry. As a proof of concept, the HER2 receptor was targeted by a nanobody anti-HER2 fused to the E3 peptide (hereinafter referred to as nano-HER2-E3) co-assembled with either the labeled synthetic K3 peptide or with the K3 peptide genetically fused to various FPs and expressed as recombinant proteins. Our data clearly demonstrate that this non-covalent labeling strategy allows the formation of robust and modular site-specific labeled probes for immuno-detection of receptor proteins in fluorescence microscopy or flow cytometry.

## 2. Results

The bivalent nano-HER2-E3, already described in our previous studies [16], was evaluated as a molecular probe for imaging and flow cytometry. Conjugation of the nano-HER2 with Alexa-Fluor-488 dye via lysine residues using *N*-hydroxysuccinimide (NHS) esters has been shown to result in a strong affinity drop of 20 to 1000 fold [16,19]. Thus, as the nanobody contains two cysteine residues and its oxidative status appeared not to be critical for epitope recognition [20], we tried to perform non-specific labeling on cysteine residues using maleimide conjugation. However, the degree of labeling was extremely low and could not be implemented for imaging and flow cytometry.

A cysteine residue was then introduced at the C-terminal part of the nano-HER2-E3 and used as unpaired cysteine for covalent maleimide conjugation using an Alexa-Fluor-488 dye (referred to as nano-HER2-E3-C). The nano-HER2-E3-C was then produced in *E. coli*, but at lowest yields compared to the parental counterpart, as previously observed with other nanobodies [2]. Introduction of a cysteine residue caused extensive dimerization as shown by the size-exclusion chromatography (Appendix A), where the nano-HER2-E3-C eluted into two peaks without a reducing agent, TCEP. Thus, reduction using TCEP was needed prior labeling with maleimide conjugate Alexa-Fluor-488 dye, but the degree of labeling was modest and never reached more than 70%, leading to a heterogeneous labeling population. 

Therefore, we decided to evaluate the co-associating tags as a labeling device. Nano-HER2-E3 was evaluated for co-assembly with various fluorescently labeled K3 peptides. To that end, the K3 peptide was either genetically fused to a C-terminal part of FPs such as eGFP or mScarlet (hereafter referred to as eGFP-K3 and mScarlet-K3, respectively) or chemically synthetized with an additional cysteine residue (hereinafter referred to as C-K3) (Figure 1). We took advantage of the absence of cysteine residue in the K3 peptidic sequence to introduce a N-terminal cysteine residue that can be selectively conjugated to maleimide-functionalized fluorochromes. 

The different recombinant proteins were produced at extremely high levels in *E. coli* and subsequently purified by immobilized metal affinity chromatography followed by size-exclusion chromatography. The final yields of all purified proteins were above 100 mg/L. Elution profiles revealed that all K3-fused displayed profiles corresponding to tetramers with a size between 155 and 160-kDa (Appendix A), whereas nano-HER2-E3 behaved as dimers, as previously described [16]. In addition, the elution profile of the different proteins on size-exclusion chromatography showed a major single peak, indicating an oligomerization of almost 100%. 

We next evaluated the ability and the stoichiometry of the complex formation between nano-HER2-E3 and its various cognate K3 partners. Following western blot transfer, the K3 moieties were detected by direct fluorescence using a Typhoon^TM^ device, whereas the nano-HER2-E3 was detected using an anti-c-myc monoclonal antibody followed by a secondary anti-mouse mAb either IR700-labeled (red) or IR800-labeled (green). Increasing amounts of purified nano-HER2-E3 were mixed with steady amounts of eGFP-K3 or Alexa-Fluor-647-labeled synthetic K3 peptide (hereinafter refereed as to 647-C-K3). Mixtures were analyzed by native polyacrylamide gel electrophoresis (Figure 2). The free eGFP-K3 (green signal) was gradually titrated by increasing amounts of nano-HER2-E3 (red signal) leading to the formation of yellow fluorescent bands corresponding to the heterotetramers (Figure 2A). At a molar ratio of 1:1, the eGFP-K3 was entirely complexed to the nano-HER2-E3. By comparison, the same experiment conducted with eGFP-E3 instead of eGFP-K3 showed no association with nano-HER2-E3 (Figure 2B). As expected, the free homodimer eGFP-E3 migrated faster in the native gel electrophoresis than the homotetramer eGFP-K3. Likewise, the 647-C-K3 synthetic peptide (red signal) displayed a strong size shift in native polyacrylamide gel electrophoresis subsequent to its co-association with the nano-HER2-E3 (green signal) (Figure 2C). Furthermore, eGFP-K3 was entirely titrated when mixed with an excess of mCherry-E3 in 85% fetal bovine serum (FBS), confirming a strong and highly specific interaction between E3 and K3 peptides independently of the fusion partner (Appendix A). Thus, the visualization of these complexes clearly demonstrates that the E3 peptide efficiently co-associates with its K3 cognate peptide partner fused to the eGFP or labeled with Alexa-Fluor-647 dye.

We confirmed the efficiency of the heterotetramerization by analytical size-exclusion chromatography (SEC) (Appendix A). When equimolar amounts of nano-HER2-E3 and eGFP-K3 were mixed and analyzed by SEC, new complexes were observed in both cases. These new complexes showed intermediary molecular weights compatible with the theoretical molecular weight of the heterotetramer (Appendix A). In addition, elution profiles established that the homotetramer eGFP-K3 dissociates to the benefit of more stable heterotetrameric complexes formed with nano-HER2-E3. For the heterotetramer nano-HER2-E3/647-C-K3, the SEC was not resolutive enough and only a very slight difference in molecular weight could be observed (Appendix A). However, a SEC could clearly separate nano-HER2-E3/647-C-K3 from the free peptide 647-C-K3. Overall, the formation of the complexes was up to 95%, as no residual homo-oligomers could be detected.

After the initial in vitro validation, we decided to evaluate the close interaction between E3 and K3 moieties at the cell surface by using Försters resonance energy transfer (FRET) (Figure 3). As energy transfer in FRET requires proximity of less than 10 nm between the donor and the acceptor, this assay can confirm the association between the nano-HER2-E3 and the various fluorescent K3 derivatives. Two fluorophores suitable for FRET were chosen to perform flow cytometry, with Alexa-Fluor-488 as the donor and mScarlet as the acceptor (Figure 3A). The nano-HER2-E3-C-488 co-associated with mScarlet-K3 was tested by flow-cytometry on two breast cancer cells lines, HER2-positive HCC1954, HER2-negative MDA-MB-231 [21] and on HER2-silenced HCC1954 cells using small-interfering oligonucleotides (Figure 3A,B and Appendix A). The fluorescent moieties were excited at 488 nm, and the signal was recorded through their respective filters, 533/30 for Alexa-Fluor-488 and 585/40 for mScarlet. As expected, a clear signal was detected around 525 nm on HER2+ HCC1954 cells incubated with nano-HER2-E3-C-488 with a mean fluorescence intensity (MFI) of 5258 (Figure 3B), whereas no detectable signal was observed on HER2- MDA-MB-231 or on HER2-silenced HCC1954 cells under the same conditions (Appendix A). For the nano-HER2-E3/mScarlet-K3 complex, only a weak signal was observed on HCC1954 cells (MFI of 351), as the mScarlet is weakly excited at 488 nm (Figure 3A,B). However, when the two fluorophores were paired in the nano-HER2-E3-C-488/mScarlet-K3 complex, a clear increase of the mScarlet signal intensity could be observed on HCC1954 cells. The mScarlet MFI was 982 and was accompanied with a decrease in the Alexa-Fluor-488 signal intensity (MFI of 3431), as expected following energy transfer from the donor to the acceptor. These observations indicate that FRET is occurring within the nano-HER2-E3-C-488/mScarlet-K3 complex, confirming the efficient co-association between nano-HER2-E3-C-488 and mScarlet-K3 at the cell surface. 

In order to validate the results of our flow cytometry-based FRET assay in an independent setting, we performed confocal microscopy. Heterotetrameric complexes with distinct fluorophore pairs, nano-HER2-E3-C-488 co-assembled with mScarlet-K3, were incubated either directly in the culture media of the HCC1954 cells for 30 min at 37 °C or after fixation of the cells. In both cases, 100% of the cells showed an intense and distinguishable staining at the cell membrane and displayed a high amount of color-coded FRET (Figure 3C), confirming the results obtained with our FACS-based FRET assay (Figure 3A,B) and clearly demonstrating the close proximity of E3 and K3 partners. In addition, the strong FRET signal on living cells experiment proves the stability of the E3-K3 interaction even in culture media. No signal could be detected on MDA-MD-231 control cells, nor with nano-HER2-E3-C-488 or nano-HER2-E3/mScarlet-K3 alone, confirming the specificity of the signal (Appendix A). Similar results were obtained with the fluorophore pairs, nano-HER2-E3-C-568 co-assembled with eGFP-K3 (Appendix A). After this validation, we wanted to determine if our labeling strategy could be comparable and therefore potentially substitute the classical indirect immunofluorescence protocol. Nano-HER2-E3 co-associated with either 488-C-K3 or eGFP-K3 was compared to an anti-HER2 mouse mAb, followed by a secondary Alexa-Fluor-488 mAb. No obvious differences could be observed between both methods (Appendix A).

Finally, we wanted to assess whether the formation of these complexes via E3 and K3 peptides affected the binding efficiency of the nano-HER2 at the cell surface HER2 receptor. To that end, the relative binding constants of the nano-HER2-E3, either alone or co-associated with the synthetic peptide 488-C-K3 or the eGFP-K3, were assessed on HER2^+^ HCC1954 and HER2^-^ MDA-MB-231 breast cancer cells by flow cytometry (Figure 4). As shown in Figure 4B, association of either eGFP-K3 or K3-C-488 with nano-HER2-E3 led to only a five-fold decrease in the relative binding affinity compared to nano-HER2-E3-C-488. Indeed, the relative binding efficiency was 6.3 ± 1.8 nM for nano-HER2-E3-C-488, as previously estimated [16]; 24.3 ± 7.0 nM for the complex nano-HER2-E3/eGFP-K3; and 28.7 ± 8.1 nM for the nano-HER2-E3/K3-C-488 heterotetramer. No signals were observed on HCC1954 cells incubated with nano-HER2/eGFP-K3, confirming the strict dependency between the fluorescent positive signal and the co-association of nano-HER2 with fluorophore via the E3 and K3 peptides (Figure 4B). Moreover, no signals were detected on HCC1954 (HER2^+^) cells incubated with a mock nanobody directed against the GFP protein (nano-eGFP-E3), described elsewhere [18], or on MDA-MB-231 (HER2^-^) cells, demonstrating the binding efficiency and specificity of the nano-HER2-E3 and fluorescent K3 complex for the HER2 cell surface receptor.

## 3. Discussion

Numerous labeling methods have been developed to conjugate fluorochromes or fluorescent proteins to antibody fragments, such as scFv or V_H_Hs [13,22,23]. Conventional bioconjugation of an organic fluorophore to lysine or cysteine residues of an antibody or derivatives suffers several drawbacks. First, an indiscriminate coupling reaction with respect to the target amino acid residue can lead to partial or even complete disruption the antigen-binding site, resulting in the generation of non-functional probes. In addition, chemical conjugations generally result in a heterogeneous mixture of antibody fragments having different number of fluorochromes per molecule. Thus, defining the coupling site is a key goal to overcome these limitations [6].

Here, we described a novel peptide tag-based labeling method taking advantage of a strong, non-covalent, highly specific conjugation between two short co-associating peptides, E3 and K3, for site specific labeling. In our approach, the E3 tag was introduced at the C-terminal end of the anti-HER2 V_H_H, giving rise to nano-HER2-E3, and the K3 peptide was either chemically synthetized with a modifiable amino acid or genetically fused to the N-terminal of various FPs. By separating the molecular probe into two distinct entities, our system is able to overcome yield, folding and solubility issues, which can be major problems when producing chimera-encompassing FPs in bacteria. For instance, the chromobody V_H_H-CD38-eGFP expressed in *E.coli* was mostly soluble, but with a yield of only around 6 mg of pure recombinant protein per liter of culture [14]. In our case, the strategy to separate both entities is illustrated by the excellent yields of the production: up to 100 mg per liter for the V_H_H and more than 120 mg per liter of culture for both eGFP-K3 and mScarlet-K3.

Then, association of both partners takes place in few minutes at RT or on ice, in a high concentration of serum and at nanomolar concentration. Unlike metal-ion peptide tags, no apparent toxicity and unspecific binding were noticed with our system. This technique is particularly straightforward and convenient, and co-assembly occurs efficiently with different fluorescent partners, such as Alexa-Fluor maleimide K3 peptide or FPs genetically fused to K3 peptide.

In addition, this technology allows a great flexibility in the choice of the fluorescent partner to be associated. Several fluorescent K3 probes were tested in association with nano-HER2-E3 without loss of binding specificity and efficiency. An indirect advantage of the system is that bivalency of the nano-HER2 is also achieved by the dimerization of the E3 tag, improving its functional affinity compared to the monovalent counterparts, by mean of avidity [18].

Analysis of the association between the V_H_H fused to E3 and the K3 derivate-labeled peptides demonstrate that pairing preferences led exclusively to the rapid formation of heterotetrameric complexes to the disadvantage of the homodimer and homotetramer monovalent counterparts. Complexes were remarkably stable during gel electrophoresis and in size-exclusion chromatography. Using FACS and confocal fluorescent microscopy, we showed that our bio-conjugated probes are highly stable in culture media to stain specifically the membrane of HER2 positive cells. In confocal experiments, the signal given by the various probes incubated before fixation of the cells was comparable to the signal detected after incubation of the probes on fixed cells. These results were further corroborated with the strong FRET signal, demonstrating that at least at the cell membrane E3 and K3 are tightly associated. Another advantage of these molecular probes is to avoid the use of secondary labeled antibodies and extensive washing steps, simplifying the procedure and reducing potential background signal. Finally, the nano-HER2 heterotetramers demonstrated equivalent apparent binding efficiency on HCC1954 cells as the nano-HER2 homodimer, demonstrating that co-association of both partners does not affect the folding and/or the binding efficiency.

In conclusion, we have developed a highly efficient and specific tool for site-specific labeling of nanobodies resulting in a homogeneous tracer population. Our system can be seen as a “two-in-one” modular site-specific labeling system. It takes advantage of the dimerization E3 tag to increase the affinity of the nanobody by means of avidity, and uses it as an anchor for its co-associating partner, K3, for a labeling device. Moreover, connecting E3 and K3 peptides to macromolecules does not compromise the activity of the fusion partners and can be easily achieved by genetic engineering or chemically. This straightforward and inexpensive technology can be scaled up for mass production of nanobody-based reagents. Finally, tethering two distinct modular entities into molecular probes allows a great flexibility in terms of co-associating partners. Altogether, E3 and K3 tags represent a versatile methodology for conjugating macromolecular probes or different chemical moieties to a protein of interest.

## 4. Materials and Methods

### 4.1. Cell Lines 

HCC1954 and MDA-MB-231 cell lines were maintained as monolayers in Roswell Park Memorial Institute-1640 medium (RPMI-1640; Life Technologies, Carlsbad, CA, USA) without HEPES, supplemented with 10 % fetal calf serum (FCS) and Gentamycin (50 µg/mL). Cells were cultivated at 37 °C with 5% CO_2_ in a humidified atmosphere.

### 4.2. Plasmids

Nano-HER2-E3 expression plasmid has been described elsewhere [16,18]. The K3 coding sequence was introduced into the NheI-EcoRI restriction sites of pET-eGFP-E3 to replace the E3 sequence, leading to pET-eGFP-K3 [15]. A sequence encoding for mScarlet and optimized for *E. coli* expression was synthetized by Integrated DNA Technologies (IDT), amplified by PCR and cloned into NcoI-NheI restriction sites in frame with K3 sequence of the pET-eGFP-K3.

### 4.3. Expression and Purification of the Recombinant Fusion Proteins

Briefly, E3 and K3 fusion proteins were overexpressed in *E. coli* BL21 (DE3) pLysS after induction with 0.5 mM isopropyl thiogalactoside (IPTG). After 24 h at 20 °C, the cells were harvested, re-suspended in 20 mM Phosphate buffer pH 8, 250 mM NaCl and 10 mM imidazole. Following lysis with lysozyme and sonication, cell debris was removed by ultra-centrifugation at 36,000× *g*, and the supernatant was applied to IMAC chromatography column charged with cobalt (GE Healthcare Saclay, Paris, France). IMAC purified fractions were subsequently loaded on a HiLoad 16/60 Superdex 200 prep grade column (GE Healthcare, Bio-sciences AB, Uppsala, Sweden) operating at a flow rate of 0.5 mL/ min. Aliquots of the fractions were separated on SDS-PAGE gels and analyzed by Coomassie blue staining. Complex formation between labeled recombinant proteins was controlled by analytical chromatography on Superdex 200 Increased 5/150 GL (GE Healthcare, Bio-sciences AB, Sweden).

### 4.4. Protein and Peptide Labeling

The peptide K3: CALNNGEYFTLQIRGRERFEMFRKLNKALELKDAQA synthetized by GeneCust (Luxemburg) was freshly dissolved in water before use. Nano-HER2-E3-C and the C-K3 peptide were labeled on a cysteine residue with maleimide Alexa-488, Alexa-567 or Alexa-647, as previously described [24]. 

### 4.5. Transient siRNA Transfections

Transient siRNA transfections were performed using Lipofectamine RNAiMAX (Invitrogen, P/N 56532, Carlsbad, CA, USA) according to the manufacturer’s instructions. HER2-targeting siRNAs were SMARTpoolON-TARGETplus obtained from Dharmacon. For controls, siRNAs ON-TARGET plus non-targeting pool from Dharmacon were used. siRNAs were used at a final concentration of 10 nM, and cells were transfected 24–72 h prior to experiments. 

### 4.6. Fluorescence-Activated Cell Sorting (FACS)

Trypsinized cells were incubated with a mixture of equimolar ratio of recombinant proteins as indicated, for 30 min at 4 °C in FACS Buffer (PBS with 0.5% BSA and 2 mM EDTA). Cells were washed twice in FACS Buffer and analyzed on an Accuri™ C6 Plus flow cytometer (BD Bioscience, San Jose, CA, USA). The relative mean fluorescence intensities were normalized and plotted against the concentration of the nano-HER2 at monomer concentration. The data shown here are single-point measurements.

### 4.7. Immunofluorescence

The cells grown on coverslips were either incubated with the labeled complex and fixed with 4% (*w*/*v*) paraformaldehyde for 30 min or directly fixed and permeabilized with 0.2% Triton X-100 for 5 min before incubation with the labeled complex or with a commercial anti-HER2 mouse mAb (9G6) was used for indirect immunofluorescence (Thermofischer, Carlsbad CA, USA). The coverslips were mounted with Fluoromount G containing 4’,6’-diamidino-2 phenyleindole (SouthernBiotech, Birmingham, UK). Images were captured with deconvolution (confocal) fluorescence microscopy (Leica DMIRBE inverted microscope and OPENLAB 3.1.4 software, Nussloch, Germany) and images were processed using Fiji. Images were filtered by applying the difference between two Gaussian Blurs (1.5 and 25).

### 4.8. Native-PAGE 

Separation of proteins by non denaturing polyacrylamide gel was performed, as previously described [15]. Shortly after this, purified complexes were mixed as indicated, and resolved on nondenaturing polyacrylamide gradient gel (8−18%) at pH 8.8. Proteins were transferred on nitrocellulose membrane and visualized using a direct fluorescence Typhoon FLA 9500 biomolecular imager (GE Healthcare) at 555 nm or detected with an anti-myc mAb and a secondary antibody IRDye800-conjugated anti-mouse (Science-Tec) and visualized on an Odyssey Classic Infrared Imager (LI-COR).

## Figures and Tables

**Figure 1 ijms-23-14405-f001:**
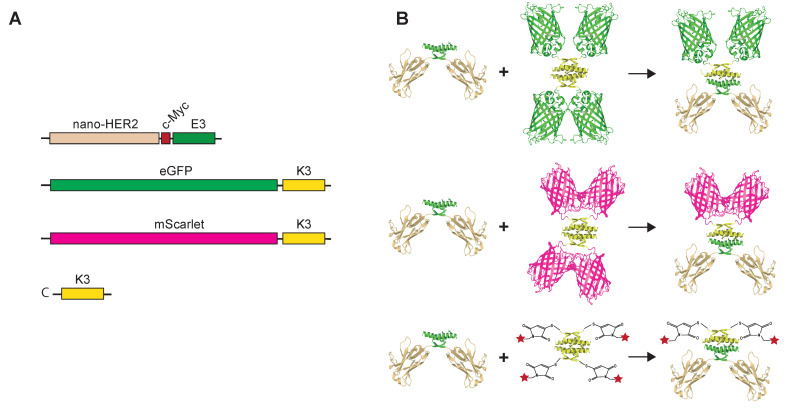
Schematic representations of the nano-HER2-E3, eGFP-K3, mScarlet-K3 and labeled-C-K3 synthetic peptides. Cartoon representing the coding sequences of each module (**A**) and the theoretical oligomerization states of the corresponding proteins (**B**) are depicted. The red star represents the Alexa-Fluor Dye.

**Figure 2 ijms-23-14405-f002:**
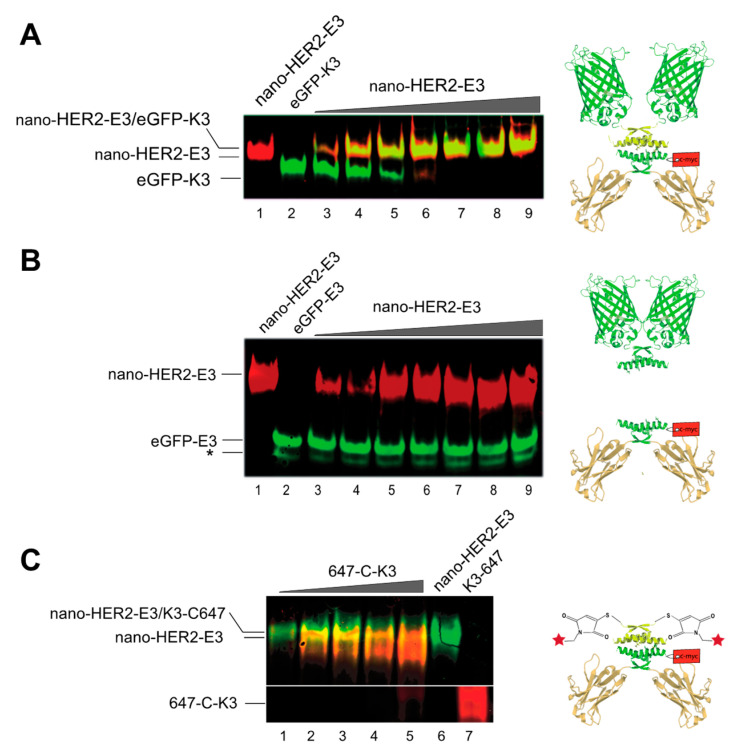
Visualization of heterotetramer formation between the nano-HER2-E3 and either eGFP-K3, eGFP-E3 as a negative control, or synthetic K3 peptide labeled with Alexa-fluor-647 (647-C-K3) in native polyacrylamide gel electrophoresis. Constant amounts of eGFP-K3 (**A**) or the eGFP-E3 (**B**) proteins were mixed with increasing amounts of nano-HER2-E3, as indicated (* degradation product of the eGFP). On the contrary, fixed amounts of nano-HER2-E3 were mixed with increasing amounts of 647-C-K3 (**C**). The nano-HER2-E3 was revealed by western blot analysis with an anti-c-Myc mAb, followed by an anti-mouse mAb either IR700-labeled (**A**,**B**) or IR800-labeled (green), and both eGFP fluorescent protein and 647-C-K3 were revealed by direct fluorescence. Molar ratios between (**A**) nano-HER2-E3 and eGFP-K3, or (**B**) nano-HER2-E3 and eGFP-E3, were as follows: 1:6 (lane 3), 1:3 (lane 4), 1:2 (lane 5), 2:3 (lane 6), 5:6 (lane 7), 1:1 (lane 8) and 7:6 (lane 9). **(C)** Molar ratios between 647-C-K3 and nano-HER2-E3 were as follows: 0:1 (lane 1), 1:10 (lane 2), 1:5 (lane 3), 3:10 (lane 4), 3:5 (lane 5) and 1:1 (lane 6).

**Figure 3 ijms-23-14405-f003:**
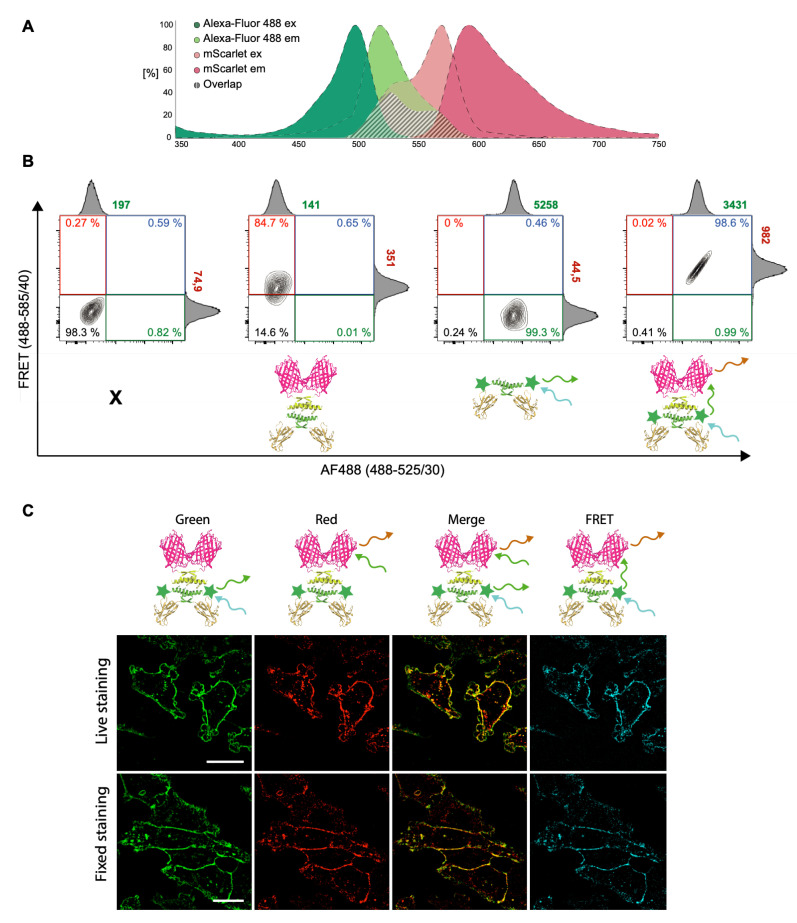
FRET between nano-HER2-E3-C-488 and mScarlet-K3 visualized by flow cytometry and by confocal microscopy. (**A**) Representation of the pair of chosen fluorophores and their respective spectra as indicated. (**B**) HCC1954 cells were incubated with the indicated complexes and analyzed by flow cytometry following excitation with a 488 nm laser. (**C**) Living fixed HCC1954 cells were incubated with the different complexes. Images were taken by confocal microscopy and analyzed with Image J. Cartoons represent the complexes. Scale bar represents 10 µm. Representative results of three independent experiments.

**Figure 4 ijms-23-14405-f004:**
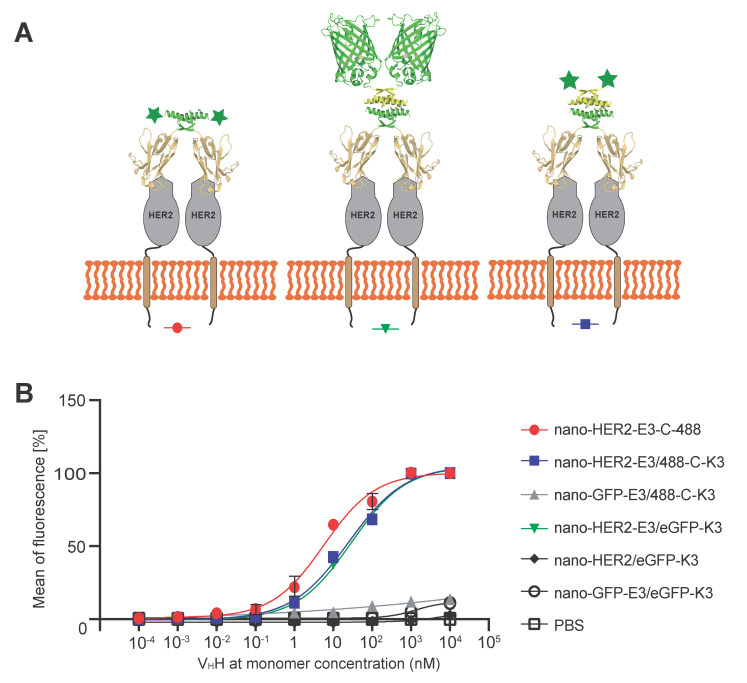
Determination of binding capacity of recombinant nano-HER2-E3 associated with eGFP-K3 or 488-C-K3 on HER2 overexpressing cells. (**A**) Cartoon representation of the heterotetramer biomolecule composed of nano-HER2-E3/eGFP-K3 binding to HER2 cell surface receptor and the positive control nano-HER2-E3-C covalently labeled with Alexa-Fluor-488 maleimide. (**B**) Determination of relative binding affinities of nano-HER2-E3 alone or either co-associated with eGFP-K3 or 488-C-K3 on breast cancer cells. HCC1954 cells were incubated with increasing concentrations of nano-HER2-E3-C-488, nano-HER2-E3/eGFP-K3 or nano-HER2-E3/488-C-K3, or with eGFP-K3 alone as a control. Following incubation, fluorescence was measured by flow cytometry. The relative mean fluorescence was plotted against nanobody concentration (nM), and the apparent K_D_ value was determined using sigmoidal fitting with R software (DRC package, R Core Team GNU GPLu2 version 4.2.1 Columbia, SC, USA).

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
