# Peer review of "Modular Site-Specific Conjugation of Nanobodies Using Two Co-Associating Tags"

_ijms, 2022, doi:10.3390/ijms232214405_

Round 1
Reviewer 1 Report
Versatile and quick site-specific conjugation of nanobodies
Summary: The authors describe the application of novel fluorescent fusion proteins and the E3/K3 peptide hetero-dimerization axis to label a model nanobody against Her2. The manuscript is well written, and the presented data supports conclusions for the most part. However, a major revision is necessary before evaluation for publication due to the following concerns:
Major comments:
1) Introduction lacks specific goals for the manuscript. What is the purpose and what are the end goals?
2) Revise the title of manuscript: The current title points towards a specific chemical conjugation, however what’s being presented is mediated by peptide dimerization. It is not quick either as it requires gene synthesis and recombinant protein purification of E3 fused target.
3) SI is missing, although significant part of results section points to SI
4) Include a table with all fusion proteins, expected MWs, expected oligomerization state, calculated MWs from SEC
5) Include all gene and protein sequences in SI
6) Lines 116-121 (figure 2): how is nano-Her2-E3 giving a red signal? Is it fluorophore conjugated? Clarify in text
7) The manuscript covers multiple experiments to show E3/K3 dimerization, however it fails to show how the labeling strategy presented here is an improvement over the current state of technology. If the claim is that non-specific labeling with lysine/cysteine dye chemistry can cause aggregation, loss of binding etc., include a control arm and additional data showing this.
8) Figure 4: It appears like nano-Her2-E3-C-488 is the simplest labeling technique while preserving nanobody affinity. Taking this data into consideration, justify the need to evaluate other alternatives.
Overall, the manuscript suffers from low impact, the goals are not clear and the experiments support probe dimerization but do not demonstrate utility of tools presented.
Author Response
Please see the attachment document.
Major comments:
1) Introduction lacks specific goals for the manuscript. What is the purpose and what are the end goals?
As requested, we have added more details in the introduction “non-covalent labeling strategy allows the formation of robust and modular site-specific labeled probes for immuno-detection of receptor proteins in fluorescence microscopy or flow cytometry.”
2) Revise the title of manuscript: The current title points towards a specific chemical conjugation, however what’s being presented is mediated by peptide dimerization. It is not quick either as it requires gene synthesis and recombinant protein purification of E3 fused target.
We agree: the new title is now : Modular site-specific conjugation of nanobodies using two co-associating tags
3) SI is missing, although significant part of results section points to SI.
We are very sorry that the SI was not available for the reviewer 1. We included it in the PDF version.
4) Include a table with all fusion proteins, expected MWs, expected oligomerization state, calculated MWs from SEC
This table was available in the SI. We have added, as requested the calculated MWs from the SEC analysis.
5) Include all gene and protein sequences in SI
All gene and protein sequences have been included in SI
6) Lines 116-121 (figure 2): how is nano-Her2-E3 giving a red signal? Is it fluorophore conjugated? Clarify in text.
The nano-HER2-E3 contains a c-myc tag. Following native gel electrophoresis and western blotting, the nano-HER2-E3 is detected with a secondary anti-mouse mAb either IR700-labeled (red for panel A and B) or IR800-labeled (green for panel C). To clarify this experiment, additional explanation has been added in the main text and figure legend.
7) The manuscript covers multiple experiments to show E3/K3 dimerization, however it fails to show how the labeling strategy presented here is an improvement over the current state of technology. If the claim is that non-specific labeling with lysine/cysteine dye chemistry can cause aggregation, loss of binding etc., include a control arm and additional data showing this.
We have explained more in details why we have been developing the E3-K3 system and added these two paragraphs in the results:
“The bivalent nano-HER2-E3, already described in our previous studies[16] was evaluated as a molecular probe for imaging and flow cytometry. Conjugation of the nano-HER2 with Alexa-Fluor488 dye via lysine residues using N-hydroxysuccinimide (NHS) esters has been shown to result in a strong affinity drop of more than 20- to 1000-fold[16,19]. Thus, as the nanobody contains two cysteine residues and it oxidative statue appeared not to be critical for epitope recognition[20], we tried to perform non-specific labeling on cysteine residues using maleimide conjugation. However, the degree of labeling was extremely low and could not be implemented for imaging and flow cytometry (data not shown).
A cysteine residue was then introduced at the C-terminal part of the nano-HER2-E3 and used as unpaired cysteine for covalent maleimide conjugation with an Alexa-Fluor-488 dye (referred to as nano-HER2-E3-C). The nano-HER2-E3-C was produced in E. coli at lowest yields compared to the parental counterpart, as previously observed on other nanobodies[2]. Introduction of a cysteine residue caused extensive dimerization as shown by last purification step by size exclusion chromatography analysis (Figure S1A and B), where the nano-HER2-E3-C eluted into two peaks without a reducing agent such as TCEP. Finally, reduction using TCEP was needed prior labeling with maleimide conjugate Alexa-Fluor-488 dye, but the degree of labeling was modest and never reached more than 70% leading to heterogeneous labeling population. “
8) Figure 4: It appears like nano-Her2-E3-C-488 is the simplest labeling technique while preserving nanobody affinity. Taking this data into consideration, justify the need to evaluate other alternatives.
As explain above, the C-terminal cysteine can be problematic in term of production and stability of the nanobody. In addition, it has to be tested for each particular nanobody.
We have added the following sentence in the conclusion “Our system can be seen as a “two-in-one” modular site-specific labeling system. It takes advantage of the dimerization E3 tag to increase the affinity of the nanobody by means of avidity, and using it as an anchor for its co-associating partner K3 for labeling device. »

Author Response
Please see also the PDF
Reviewer 2
1) In the figure 2, could you explain why there is such a high difference of migration between the panel A and B. Indeed, the distance of migration between the nano-HER2-E3 and the eGFP-E3 is almost twice the one between the nano-HER2-E3 and the eGFP-K3 while the size of the peptides is supposed to be the same. Is this just a difference of time of migration, and if it is, can you add the size of the markers on the side.
Effectively, the distance in migration of eGFP-E3 is twice the one of eGFP-K3 because the E3 helix forms only homodimer, while K3 forms homotetramer. As it was not very well explained in the main text, we added an additional sentence.
2) While the different components are efficiently associated in vitro and seems to interact in cellulo at the surface at the cell (FRET experiments), the question of the specificity of recognition of the HER2 receptor could be improved. Could you first add a western-blotting showing the absence of expression of HER2 in the MDA cell line compared to the HCC cell line, or at least a reference showing this result?
As requested, we have added a reference showing the absence of HER2 expression on MDA-MB231 triple negative breast cancer cell line and the HER2-positive HCC1954 breast cancer cell line. Kataoka, Y.; Mukohara, T.; Shimada, H.; Saijo, N.; Hirai, M.; Minami, H. Association between Gain-of-Function Mutations in PIK3CA and Resistance to HER2-Targeted Agents in HER2-Amplified Breast Cancer Cell Lines. Ann. Oncol. 2010, 21, 255–262, doi:10.1093/annonc/mdp304.
It would also be interesting to test your complexes on one other positive cell line, and one negative, for the expression of the HER2 receptor. Even better would be the use of a siRNA against the HER2 transcript in the HCC cell line to show a decrease of staining by your nanobodies after its invalidation.
As requested, we perfomed a FACS experiment using siRNA directed against HER2 transcripts in HCC1954 to show the specificity. This experiment has been added in the supplementary file (Figure S4).
3) What about the stability of the complexes in the medium or inside the cells? The in cellulo experiments were performed with a 30 min incubation at 4°C, at least for the cytometry. So, could it be possible to increase the time and do it at 37°C in a serum-containing medium to quantify the stability of these complexes overtime.
For the cytometry experiment, the experiment was performed at 4°C during 30 min, however for the confocal, we performed in parallel the incubation at 37°C on the cultured living cells in the incubator and compare the staining with a conventional IF (fixation of the cells followed by incubation with the Ab).
As we could not see any major differences in staining and intensity between both experimental protocols, we concluded on the stability of these complexes in serum-containing media. In addition, the E3-K3 complex could form in the presence of 85% serum (SI). (figure S2)
4) Regarding the confocal microscopy experiments, are 100% of the cells stained with the nanobodies? If one of the goals of your nanobodies is to replace the indirect immunofluorescence protocol, could you compare the staining obtained with your nanobodies to the one obtained with a commercial anti-HER2 (and a secondary one).
The experiment has been performed and included in the supplementary information
Yes, 100% of the cells are stained using the nano-HER2-E3 paired with either 488-C-K3 or eGFP-K3. We have added an experiment (classical immunofluorescence) proving this issue in a new supplementary (figure S6)
As requested, we have compared the signal obtain in IF with our technology and with a conventional mAb followed by a secondary Alexa488 labelled mAb. Both technics are equivalent, but with the nano-HER2-E3, the complex is preformed with either 488-C-K3 or eGFP-K3 and it is faster, because no secondary incubation is needed.
However, the eGFP is not as bright as the Alexa-488. Thus, the two technics indistinguishable in term of bright staining are nano-HER2-E3 complexed with 488-C-K3 and the mAb anti-HER2 followed by an Alexa-fluor 488 labelled secondary Ab.
5) In the discussion, it is stated that these complexes could be used to follow the internalization of the receptors and are stable in the endosomes but where are the data or the references supporting this statement? For me, the confocal microscopy images cannot lead to this conclusion.
We agree with the reviewer that the confocal images cannot support our previous statement that the E3-K3 interaction is stable in the endosome. As a consequence, we have removed the corresponding statement.
Minor comments:
- 1) The confocal microscopy images might be improved since they are not really bright on the pdf. The confocal images have been improved.
- 2) There is one reference missing in the discussion, page 8. As we have reorganized the conclusion, this issue is no longer of concern.

Round 2
Reviewer 1 Report
The authors have addressed my concerns. I believe the manuscript can be published in the current state.
Reviewer 2 Report
I thank the reviewers for clearly answering my questions and to add all the needed information to clarify their data and conclusions. I therefore think that this manuscript can be published in the present form.